# YOLOR-BASED MULTI-TASK LEARNING

## ABSTRACT

Multi-task learning (MTL) aims to learn multiple tasks using a single model and jointly improve all of them assuming generalization and shared semantics. Reducing conflicts between tasks during joint learning is difficult and generally requires careful network design and extremely large models. We propose building on You Only Learn One Representation (YOLOR) (Wang et al., 2023c), a network architecture specifically designed for multitasking. YOLOR leverages both explicit and implicit knowledge, from data observations and learned latents, respectively, to improve a shared representation while minimizing the number of training parameters. However, YOLOR and its follow-up, YOLOv7 (Wang et al., 2023a), only trained two tasks at once. In this paper, we jointly train object detection, instance segmentation, semantic segmentation, and image captioning. We analyze tradeoffs and attempt to maximize sharing of semantic information. Through our architecture and training strategies, we find that our method achieves competitive performance on all tasks while maintaining a low parameter count and without any pre-training. We will release code soon.

## 1 INTRODUCTION

Multi-Task Learning (MTL) has been one of the most popular research topics in recent years. Its purpose is to simulate the ability of humans to learn multiple tasks at the same time, and to share the learned knowledge with other tasks. Using a single model to perform multiple tasks is one of the core concepts of artificial general intelligence (AGI), but such a model must also have the ability to establish correlations between different tasks, just like the human cognitive system. Therefore, it is well-founded and necessary to use MTL to connect tasks in the field of artificial intelligence (Crawshaw, 2020; Li et al., 2019; Ben-David & Schuller, 2003).

Specifically, we focus on combining multiple vision tasks. Visual tasks, such as semantic segmentation and object detection, achieve different purposes, but are similar in that they should semantically have the same definition for the object: "car," so that the correlation between tasks can be described with the correct semantics. When the correlation between tasks is properly defined, the developed system will be robust and easy to use. We intend to find tasks suitable for learning together by analyzing task correlation between object detection, instance segmentation, semantic segmentation, and image captioning, as well as maximize the semantic range that can be shared.

To demonstrate the power of MTL, we use the MTL network YOLOR (Wang et al., 2023c) and the network architecture ELAN (Wang et al., 2023b) that optimizes the transmission of gradient flow. YOLOR leverages both "explicit" and "implicit" knowledge. The former is obtained through typical supervised learning and by extracting features from data observations. The latter is stored as latents. The same semantics from different tasks should map to the same latents and allow the model to capture unified representations for MTL. However, the multitasking capabilities of YOLOR and its follow-up, YOLOv7, have not been extensively experimented with. For instance, YOLOv7 only jointly trained object detection and instance segmentation, or object detection and pose estimation. We build on YOLOR and design training strategies to maximizing shared semantic information across tasks.

The main function of ELAN is to automatically assign different information to different parameters for learning, and thereby optimize the parameter usage efficiency of the backbone. This design allows us to focus on designing a method that enables the prediction head of each task to effectively share information without needing to consider the learning mechanism of the backbone.

In addition, we observed that different data augmentation methods will damage the semantics of different tasks to varying degrees, so we propose to design the training process from a semantic perspective. We propose an asymmetric data augmentation method to reduce the negative impact caused by semantic inaccuracy, and we found that this significantly assisted MTL of visual-language (VL) and visual tasks. We also analyze and discuss the optimizer of the VL model since it includes a text decoder, which is quite different from the other heads (Xu et al., 2022).

The main contributions of this paper are summarized as follows:

- From the perspective of human vision, we develop a MTL architecture and method to maximize the shared semantics between multiple visual and VL tasks.
- We analyze data augmentation techniques and design training flow from the perspective of semantic consistency to reduce conflicts between different tasks and make training robust.
- Through extensive experiments, we find that all tasks improve through joint learning, and we achieve competitive results with state-of-the-art while maintaining a low parameter count and without any pre-training.

## 2 RELATED WORKS

### 2.1 MULTI-TASK LEARNING

The main purpose of MTL is to use a single model for learning multiple tasks, by learning multiple related tasks at the same time, so that they can assist each other when performing tasks. The learning mechanism can reduce the occurrence of overfitting by sharing parameters and improve the efficiency of learning (Caruana, 1997; Crawshaw, 2020; Zhang & Yang, 2017; Heskes, 1998). There are two common MTL models (Ruder, 2017): (1) Hard parameter sharing: This learning architecture allows all tasks to share the entire network architecture and hidden layers, and the only difference is that each task has its own output layer (i.e., decision head). For example, Ranjan et al. (Ranjan et al., 2016) put all tasks through the same backbone network, make its parameters shared, and design a output head for each task; (2) Soft parameter sharing: This architecture is different from the aforementioned hard parameter sharing. It allows each task to have its own model and normalizes the distance between model parameters, so as to regularize the use of similar parameters for different task models. Studies using this mechanism are as follows. Duong et al. (Duong et al., 2015) use $l_2$ norm, while Yang and Hospedales (Yang & Hospedales, 2016) use trace norm to perform regularization procedures respectively. Pix2Seq v2 (Chen et al., 2022) uses text tokens to output the results of all tasks (e.g., object detection, instance segmentation, pose estimation, and image captioning), and then outputs and visualizes through decoding/detokenizing post-processing. Although this mechanism unifies the output method, it still needs to define the output format of various tasks in advance and convert it back to a visual form through post-processing, so it is not intuitive from a human's point of view. Therefore, in this paper, we use hard parameter sharing and lightweight heads to output semantic features catered to each task.

The correlation of tasks is also a very important part of MTL (Crawshaw, 2020; Li et al., 2019; Ben-David & Schuller, 2003), because when the nature of the tasks is different, in addition to causing low training efficiency, it will also cause mutual interference between tasks. The speed and effect of training can be effectively improved by grouping tasks, but the analysis and grouping of tasks are very time consuming and expensive (Crawshaw, 2020; Fifty et al., 2021). Thus, we select MTL tasks from the perspective of human vision on the correlation between tasks. At the same time, we maximize shared semantics between tasks.

### 2.2 IMAGE CAPTIONING

Image captioning mainly uses natural language to describe the contents of images (Chen et al., 2015). The main body of this system is divided into two parts: one is the image encoder which encodes the input image, and the other is the text decoder which "understands" the image encoded by the image encoder and outputs it in text. Since the text decoder needs to describe the content in the image, it needs to obtain relevant content from the image encoder, such as the objects and backgrounds in the image. In order to obtain the semantics that can be used in image captioning, many methods start with object detection and semantic segmentation as shown in Figure 1. For example, Li et al. (Li

et al., 2020) use R-CNN (Girshick et al., 2014) as the object detector to detect target objects. Zhang et al. (Zhang et al., 2021) use ResNeXt (Xie et al., 2016) to detect objects, backgrounds, and their properties. In (Cai et al., 2020), Cai et al. propose to separate instance and stuff through panoptic segmentation (Kirillov et al., 2018).

Systems designed to perform image captioning typically train the image encoder first and then the text decoder through fine-tuning (Vinyals et al., 2014; Xu et al., 2015). This approach aims to train the text decoder through a pre-trained image encoder and generate appropriate captions. The existing image captioning system usually puts emphasis on the text decoder during training, and the image encoder is usually trained with a smaller learning rate in order to prevent existing knowledge from being destroyed. However, the above approach will cause a semantic gap (Xu et al., 2022) between the image encoder and text decoder, limiting the learning ability of the text decoder to the image encoder. With the introduction of Transformer (Vaswani et al., 2017), the encoder and decoder can learn at the same time in the framework, which can reduce the semantic gap between the image encoder and text decoder. In view of the fact that Transformer has achieved many successful results in the field of natural language (Devlin et al., 2018; Radford et al., 2019; Brown et al., 2020; Raffel et al., 2020), many computer vision tasks have also begun to introduce a large number of Transformer architecture (Dosovitskiy et al., 2021; Liu et al., 2021). Basically, Transformer uses a large structure and a large amount of data to encode information. It also needs to perform multiple attentions to align features, which consumes a lot of computing resources and time.

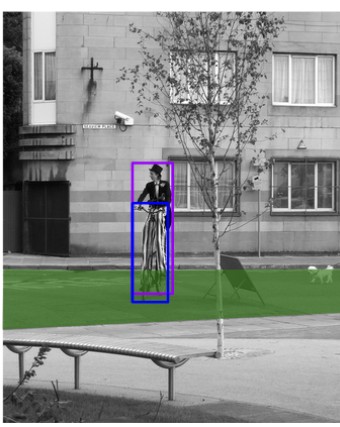

A **man** is **riding** a large **bike** through the **park**.
A **person** **riding** a very tall **bike** in the **street**.
A **man** this is on a high wheel **bicycle**.
A **person** **riding** a **bicycle** on a deserted **street**.
A **person** **rides** a **bike** on the **road**.

Figure 1: **An image captioning example** obtained by using MS COCO dataset (Lin et al., 2014). The five captions describe instances and stuff in the image, as well as the relationship between the objects and background. Purple, blue, and green text correspond to the person (purple bounding box), bicycle (blue bounding box), and segmented background (green mask) in the image respectively, while the yellow text show the relationship between objects.

## 3 ARCHITECTURE

### 3.1 OVERVIEW

In this paper, we use the ELAN design as the main axis of the network and incorporate the concept of YOLOR as the infrastructure of our system to build the backbone. Then, we adopt the hard parameter sharing mechanism for MTL, design lightweight heads for individual tasks such as object detection, instance segmentation, and semantic segmentation, and merge them into the image encoder for the image captioning task. The outputs of the heads can be used to capture features with sufficient semantics and these features can provide the need of the text encoder. Different vision tasks processed on the encoder

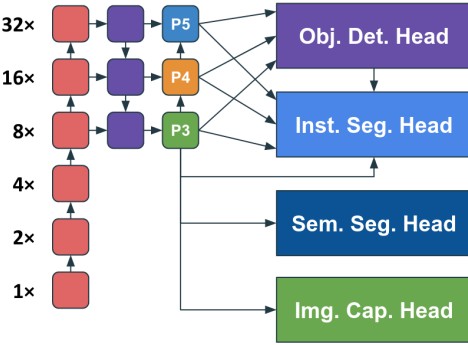

Figure 2: **Network architecture of our model.**

side can allow the system to obtain outputs with different semantics. In our proposed system, different foreground objects rely on object detection and instance segmentation to obtain accurate information, while correct background information is provided by semantic segmentation. As for the text decoder of the image captioning task, we directly use the Transformer decoder. Similar to the Transformer, we train the image encoder and text decoder together. This allows the model to be lightweight and efficient in addition to reducing training resource consumption.

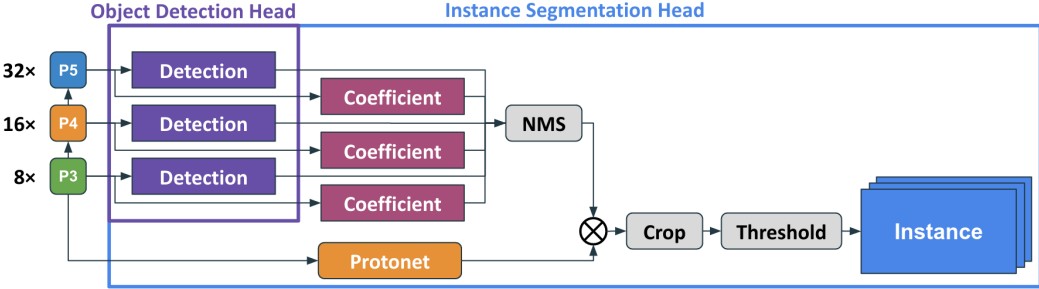

Figure 3: **Object detection head and instance segmentation head.** Instance segmentation is based on the results of object detection.

We combine YOLOR and ELAN following YOLOv7 (Wang et al., 2023a), a state-of-the-art object detection algorithm. YOLOR (Wang et al., 2023c) is a network architecture designed to perform multiple tasks. Intuitively, it mimics human learning – explicit and subconscious (implicit). Explicit learning comes from data observations and supervised training, while implicit learning comes from encoding previously learned common features and connecting them across tasks. Efficient Layer Aggregation Networks (ELAN) (Wang et al., 2023b) uses gradient path analysis to design network architectures. Its most important design concept is to optimize the gradient path. This design can make the network lighter, and it can also make the network transmit gradient information more efficiently, which can speed up and improve the representation ability of the model. The overall architecture of our model is shown in Figure 2.

Below, we describe the heads used for each task.

### 3.2 Object Detection Head & Instance Segmentation Head

We use the same object detection head of YOLOv7 (Wang et al., 2023a). YOLOv7 is the state-of-the-art in real-time object detection task, so we retain this head for our object detecion task. YOLACT (Bolya et al., 2019) is an instance segmentation model which adds an instance segmentation head to the YOLO architecture. There is a branch in YOLOv7 that merges YOLACT for the instance segmentation task. We choose this head for our instance segmentation task because it achieves state-of-the-art results in real-time instance segmentation and can be simply merged to the YOLO based architecture. The architecture for the object detection head and instance segmentation head is shown in Figure 3.

### 3.3 Semantic Segmentation Head

For the semantic segmentation task, we explore the effect of using a feature combination of single-scale and multi-scale on semantic segmentation. We design two ways to obtain semantic masks for the neck. One is to directly up-sample the feature map to $1 \times 1$ resolution from the $8 \times 8$ resolution that is closest to the original resolution (single-scale), and the other is to combine the feature maps of three different resolutions ($8 \times 8$, $16 \times 16$, and $32 \times 32$) of the neck and then up-sample to $1 \times 1$ (multi-scale). See Appendix A for more details. For semantic segmentation, the spatial relationship sensitivity is relatively important, so we choose to design the semantic segmentation head in the single-scale way at the highest resolution (i.e., $8 \times 8$). However, it would be inefficient to up-sample the feature map back to its original size, so we up-sample once at the shallowest layer (from $8 \times 8$ to $4 \times 4$) compared to the instance segmentation head. From this we get the prediction for semantic segmentation (Figure 4a). The benefits obtained by the above method are shown in Table 1a. In the same case of up-sampling back to the original size ($1 \times 1$), single-scale reduces the number of parameters by 94.3% compared to multi-scale, and reduces the training time by 6.3%. The up-sampling to $4 \times 4$ reduces the training time by 84.4% compared to the direct up-sampling to $1 \times 1$, but it can maintain fewer parameters and obtain the highest accuracy.

Table 1: **Training time and parameters for various approaches.**

(a) Comparison of training time and parameters between various approaches of the semantic segmentation head. This set of experiments uses one A5000 GPU, images are uniformly scaled to $512 \times 512$, and the batch size is 4.

| Semantic approaches | #param of seg. head | Training (hr/epoch) | FWIOU$^{val}$ |
|---|---|---|---|
| Multi-Scale $(1 \times 1)$ | 778.8K | 24 | 19.79 |
| Single-Scale $(1 \times 1)$ | **44.5K** | 22.5 | 55.55 |
| Single-Scale $(4 \times 4)$ | **44.5K** | **3.5** | **56.44** |

(b) We use the CATR (Uppal & Abhiram, 2018) model as a basis to compare the difference in cost between the full Transformer and Transformer decoder as the text decoder. This set of experiments uses one A5000 GPU, images are uniformly scaled to $512 \times 512$, and the batch size is 8.

| Text Decoder | #param | Training (hr/epoch) | B@4$^{val}$ |
|---|---|---|---|
| Full Transformer | 112.77M | 4.0 | 25.3 |
| Transformer Decoder | **104.88M** | **3.0** | **25.6** |

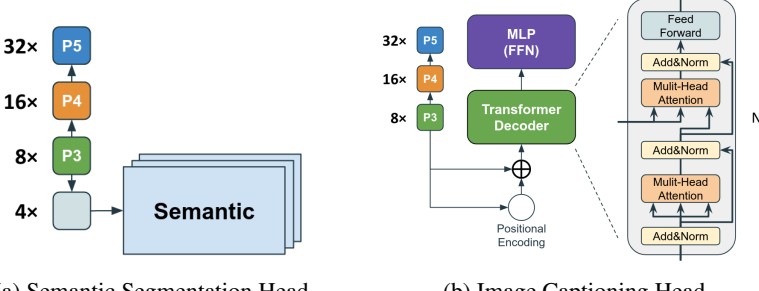

(a) Semantic Segmentation Head      (b) Image Captioning Head

Figure 4: **Heads designed for different tasks**: (a) The head designed for the semantic segmentation task. We use the feature map with the highest resolution in the neck to obtain the prediction for semantic segmentation. (b) The head that executes the image captioning task. We use the basic Transformer decoder as the text decoder that produces the captions.

## 3.4 IMAGE CAPTIONING HEAD

Following the success of Transformer (Vaswani et al., 2017) in the natural language domain, for the task of image captioning, we implement self-attention in our backbone and use it as our image encoder. Then, we incorporate a Transformer text decoder as our image captioning head. Compared to Transformer, we reduce a significant number of parameters (Figure 4b). As shown in Table 1b, we use CATR (Uppal & Abhiram, 2018), which uses the full Transformer as the text decoder, and compare the results with only using the Transformer decoder. The overall parameter count is reduced by 7.5%, the training time is reduced by 25%, and the BLEU-4 score is slightly improved. We hypothesize that the full Transformer has worse performance due to its conflicts with the backbone, as the backbone already performs self-attention, which overlaps with the function of the Transformer encoder.

Following the results of this experiment, we use a vanilla Transformer text decoder, and again use ELAN+YOLOR for the backbone.

## 4 TRAINING STRATEGIES

### 4.1 DATA AUGMENTATION

Data augmentation is used to increase the robustness of the model. However, not all data augmentation methods are necessarily valid, and some of them will cause semantic inconsistency between the data space and label space. For example, combining multiple images using the MixUp or Mosaic technique can enrich the information of the object and the background, and at the same time improve the accuracy of object detection (Bochkovskiy et al., 2020). However, for these data augmentation techniques

applied on image captioning, the additional image samples have nothing to do with the content of the original image and may even confuse the objects and background. This will cause the semantics contained in the augmented image to not correspond to the captions that have been annotated, which will lead to semantic errors. In response to this situation, we designed a very simple and intuitive training flow to learn different tasks. See Appendix B for the detailed architecture. In order to avoid the above-mentioned semantic errors, we follow the principle of "after changing the content of the original image, will the semantic information of corresponding task also be changed?" for executing data augmentation. We sort out several data augmentation pipelines according to the nature of each task. For the object detection, instance segmentation, and semantic segmentation tasks, we choose to use MixUp (Zhang et al., 2018), Cutout (DeVries & Taylor, 2017), Mosaic (Bochkovskiy et al., 2020), Copy paste, and Random perspective. For the image captioning task, we use resize and padding. We also apply different data augmentation pipelines on the same image simultaneously, and this action allows all tasks to maintain the correctness of the target semantics during the learning process.

The aforementioned design strategy can take into account the robustness of the model in visual tasks and simultaneously maintain the consistency of data and semantics between different tasks. We show the performance of the overall system in Table 2. Note that the above design concept can help the model have better multi-task scalability. Future adopters of this design concept must understand that the training time of this system is proportional to the number of data augmentation pipelines due to the shared backbone, but it does not affect the overall inference time.

Table 2: **Comparison between different data augmentation techniques.** All results are obtained by terminating at Epoch 30, training from scratch for 300 epochs.

| Augmentation | | OD | IS | StuffS | IC |
|---|---|---|---|---|---|
| Vision tasks | IC task | $AP^{val}$ | $AP^{val}$ | $FWIOU^{val}$ | $B@4^{val}$ |
| Strong | Strong | 31.4 | 25.3 | 48.1 | 7.0 |
| Strong | Weak | **35.6** | **28.8** | **53.5** | **16.2** |

Note. OD = Object Detection; IS = Instance Segmentation; StuffS = Stuff Segmentation; IC = Image Captioning.

### 4.2 OPTIMIZER STRATEGY

An image captioning model often pre-trains an image encoder, then finetunes the encoder while training a new text decoder, so it is common to give the image encoder a smaller learning rate than text decoder, which can avoid excessive interference with the knowledge that image encoder has already learned. However, we hypothesize that since we are training multiple tasks at once, the image encoder needs to be able to adapt to different data types and outputs, and therefore require a larger learning rate. We run a set of experiments to verify this. The settings of these experiments are as follows: we use AdamW and linear decay and we set the larger and the smaller learning rates to 1e-4 and 1e-5, respectively. When both sides need the same learning rate, we set it to 1e-4. We analyzed a total of three different situations, and the corresponding experimental results are shown in Figure 5. Let the learning rate of image encoder and text decoder be $l_{ie}$ and $l_{td}$ respectively. We analyze three different situations below: (1)

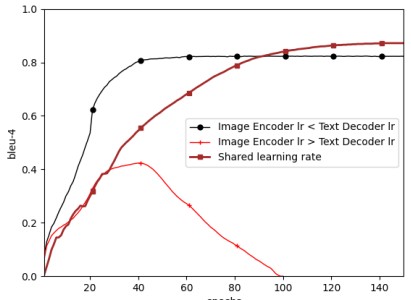

Figure 5: **Comparing the learning curves of the image encoder and text decoder on the image captioning task when using different learning rates.** The benchmark of this comparison is to use Karpathy Split validation set (Karpathy & Li, 2014) for training and validation. Although the shared learning rate mechanism takes more time to converge, because image encoder and text decoder learn together, it can easily surpass other combinations and achieve the best results.

$l_{ie} < l_{td}$: image encoder is more able to maintain pre-trained knowledge, which will lead to faster learning of the text decoder. But when the image encoder learns for a period of time, it will stop learning new knowledge earlier, and this will cause the features passed by the image encoder to the text decoder to no longer change. This situation will cause the text decoder to fail to converge to optimal results; (2) $l_{ie} > l_{td}$: a higher learning rate for the encoder results in rapid changes to its weights, while the decoder has not learned to generate sensible captions. The loss negatively affects the encoder, leading to divergence; (3) $l_{ie} = l_{td}$: although the image encoder will lose some pre-trained knowledge and make learning slow, both the image encoder and text decoder can continue to learn, and the image encoder can adapt to different data inputs and outputs, resulting in a better

learning effect than (1) and (2). From the above observation, we found that only when the pre-trained dataset has rich semantic information, the advantages brought by pre-trained knowledge can be maintained.

We transfer this understanding to our experiments where we train from scratch, and set the same learning rate for our image encoder and text decoder for optimal results.

# 5 EXPERIMENTS

## 5.1 EXPERIMENTAL SETTINGS

We use MS COCO 2017 dataset (Lin et al., 2014) for multi-task experiments. We share 118K images for training and 5K images for validation. For each image, captions are annotated by 5 people. The multitasking referred to here includes object detection, instance segmentation, semantic segmentation, and image captioning. For the semantic segmentation task, we combine the MS COCO stuff dataset (Caesar et al., 2016) and all categories of MS COCO dataset (including 80 instance categories, 91 stuff categories, and 1 unlabeled category) for training. As for verifying the effect of semantic segmentation, we use COCO-Stuff 164K dataset (Caesar et al., 2018) and COCO-Stuff 10K dataset (Caesar et al., 2018) respectively. COCO-Stuff 164K dataset has the same data as MS COCO 2017 dataset, using the same 118K images as the training set and 5K images as the testing set, while COCO-Stuff 10K dataset uses more complex 10K images from the COCO-Stuff 164K dataset, and divides them into 9K training set and 1K testing set.

Table 3: **Comparison with baseline models.** In this experiment, we use COCO-Stuff 164K dataset (Caesar et al., 2018) for comparison in the semantic segmentation task.

| model | OD AP$^{val}$ | IS AP$^{val}$ | SemS MIOU$^{test}$ | IC B@4$^{val}$ |
|---|---|---|---|---|
| YOLOv7 with YOLACT | 52.0 | **42.4** | - | - |
| YOLOv7 (Segmentation) | - | - | 37.4 | - |
| CATR | - | - | - | 26.0 |
| ours | **52.1** | **42.4** | **42.5** | **28.4** |

Note. OD = Object Detection; IS = Instance Segmentation; SemS = Semantic Segmentation; IC = Image Captioning.

## 5.2 BASELINES TO VERIFY THE EFFICACY OF MULTI-TASK LEARNING

The following models share similar architectures as ours, but are trained on fewer tasks. We verify whether jointly training our selected tasks leads to improved results.

- YOLOv7 (Wang et al., 2023a) with YOLACT (Bolya et al., 2019): object detection, instance segmentation
- YOLOv7 (Wang et al., 2023a) (Segmentation): semantic segmentation (COCO Stuff-164K)
- CATR (Uppal & Abhiram, 2018): image captioning

The version of YOLOv7 combined with YOLACT is currently the state-of-the-art of real-time object detection and real-time instance segmentation. We use this version as the basis for MTL and it is therefore adopted as the baseline for object detection and instance segmentation. In addition, we adopt YOLOv7 with a semantic segmentation head as the baseline for semantic segmentation and use it to verify the performance of semantic segmentation. CATR uses the most basic Transformer to implement image captioning. We adopt CATR, replace its image encoder with ELAN plus YOLOR, and remove its Transformer encoder to serve as the baseline of image captioning. In our experiments, we uniformly convert images to $640 \times 640$ for training and use AdamW as the optimizer. Table 3 shows the experimental results compared to the baseline. From Table 3, we know that through semantic sharing, all visual tasks achieve improved performance. It is worth mentioning that for semantic segmentation, our method significantly improves the results by 13.6% over the baseline, while image captioning improves by 9.2% compared to the baseline.

## 5.3 COMPARISON WITH STATE-OF-THE-ARTS

We compare our model with the following state-of-the-art multi-task models:

- YOLOv7 (Wang et al., 2023a): object detection, instance segmentation

Table 4: **Comparison with state-of-the-art multi-task models.**

| model | #param | image size | from scratch | OD $AP^{val}$ | IS $AP^{val}$ | IC $B@4^{val}$ |
|---|---|---|---|---|---|---|
| YOLOv7 | 34.5M | $640^2$ | ✔ | 51.4 | 41.5 | - |
| YOLOv7-AF | 45.9M | $640^2$ | ✔ | 53.0 | 43.3 | - |
| YOLOv7-AF* | 46.0M | $640^2$ | ✔ | 52.0 | 42.4 | - |
| Pix2Seq v2 | 115.2M | $640^2$ | ✘ | 44.2 | 36.9 | 34.3 |
| Pix2Seq v2 | 115.2M | $1024^2$ | ✘ | 46.5 | 38.7 | 34.9 |
| ours | 80.0M | $640^2$ | ✔ | 52.1 | 42.4 | 28.4 |

Note. OD = Object Detection; IS = Instance Segmentation; IC = Image Captioning.
★: YOLOv7 anchor free version, adopts the same settings as our method for training.

Table 5: **Comparison with state-of-the-art models.** In this experiment, we use COCO-Stuff 10K dataset (Caesar et al., 2018) for comparison in the semantic segmentation task.

| model | #param | image size | from scratch | OD $AP^{val}$ | IS $AP^{val}$ | SemS $MIOU^{test}$ |
|---|---|---|---|---|---|---|
| InternImage-T | 49M | 1280x800 | ✘ | 49.1 | 43.7 | - |
| InternImage-S | 69M | 1280x800 | ✘ | 49.7 | 44.5 | - |
| InternImage-B | 115M | 1280x800 | ✘ | 50.3 | 44.8 | - |
| InternImage-L | 277M | 1280x800 | ✘ | 56.1 | 48.5 | - |
| InternImage-XL | 387M | 1280x800 | ✘ | 56.2 | 48.8 | - |
| InternImage-H | 1.31B | $512^2$ | ✘ | - | - | 59.6 |
| ViT-Adapter-T | 28.1M | $192^2$ | ✘ | 46.0 | 41.0 | - |
| ViT-Adapter-S | 47.8M | $384^2$ | ✘ | 48.2 | 42.8 | - |
| ViT-Adapter-B | 120.2M | $768^2$ | ✘ | 49.6 | 43.6 | - |
| ViT-Adapter-L | 347.9M | $1024^2$ | ✘ | 52.1 | 46.0 | - |
| ViT-Adapter-L | 332.0M | $512^2$ | ✘ | - | - | 54.2 |
| ours | 80.0M | $640^2$ | ✔ | 52.1 | 42.4 | 50.1 |

Note. OD = Object Detection; IS = Instance Segmentation; SemS = Semantic Segmentation.

- Pix2Seq v2 (Chen et al., 2022): object detection, instance segmentation, and image captioning

- InternImage (Wang et al., 2022): object detection, instance segmentation, and semantic segmentation (COCO Stuff-10K)

- ViT-Adapter (Chen et al., 2023): object detection, instance segmentation, and semantic segmentation (COCO Stuff-10K)

We present the experimental results in Table 4 and Table 5. Experimental results show that our model is much smaller than other multi-task models and can achieve very good performance on various tasks. Among models of the same scale, our object detection result is the best. Our model is not as effective as other large models on the semantic segmentation task, but our model is very lightweight, saving an average of 75% of the parameter consumption compared to other models while achieving very good results. It is worth mentioning that all our tasks are trained together from scratch. For image captioning, we did not use any pre-trained image classifier or object detector, but let the image encoder and text decoder train together and share the semantics obtained by the tasks.

## 5.4 ABLATION STUDY

In this section we report the results of ablation study. We pair each visual task with image captioning individually. The purpose of this is to observe the amount of semantic meaning that each task can provide to image captioning. Because the result of instance segmentation depends on that of object detection, this experiment must combine the results of object detection and instance segmentation, and the overall result is listed in Table 6.

According to Table 6, semantic segmentation combined with image captioning gave the worst results. This means that the amount of semantic meaning that this combination can provide is significantly less than the combination of object detection, instance segmentation, and image captioning.

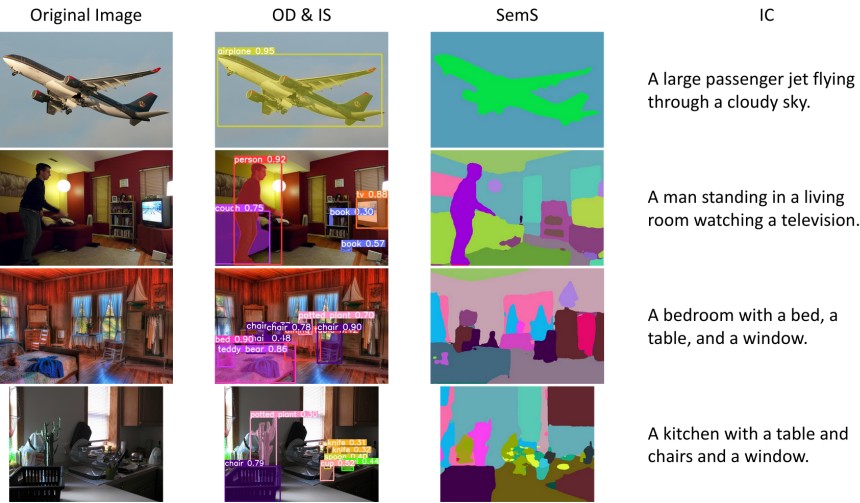

| Original Image | OD & IS | SemS | IC |
|---|---|---|---|
| | | | A large passenger jet flying through a cloudy sky. |
| | | | A man standing in a living room watching a television. |
| | | | A bedroom with a bed, a table, and a window. |
| | | | A kitchen with a table and chairs and a window. |

Figure 6: **Some results generated from our model.**

It can be found from the annotation of Figure 1 that the captioning task focuses on describing the relationship and interaction of foreground objects. For all captions in MS COCO dataset, stuff accounts for 38.2% of caption nouns (Caesar et al., 2016). When combining all tasks, since semantic segmentation cannot clearly distinguish different foreground objects, when learned together with object detection and instance segmentation, it is possible that the results of both tasks will deteriorate at the same time. But for semantic segmentation, different objects may belong to the same category. Adding object detection and instance segmentation improves the model's mask prediction and overall performance in the semantic segmentation task. The visualization of ablation study is shown in Appendix C.

Table 6: **Comparison between the pairings of image captioning and different visual tasks.** The results produced by this comparison are obtained by terminating at Epoch 20, training from scratch for 60 epochs.

| tasks | OD $AP^{val}$ | IS $AP^{val}$ | SemS $FWIOU^{val}$ | IC $B@4^{val}$ |
|---|---|---|---|---|
| OD + IC | 45.6 | - | - | 18.5 |
| OD + IS + IC | **45.8** | **37.5** | - | 20.6 |
| SemS + IC | - | - | 33.2 | 5.9 |
| All | 45.7 | 37.4 | **40.8** | **20.7** |

Note. OD = Object Detection; IS = Instance Segmentation; SemS = Semantic Segmentation; IC = Image Captioning.

### 5.5 VISUALIZATION OF EXPERIMENTAL RESULTS

We adopt the test set from MS COCO 2017 dataset (Lin et al., 2014) to demonstrate our results, as shown in Figure 6. The image captioning task is easily influenced by the misjudgement of the visual tasks, negatively affecting the content of the caption (e.g., the basket is predicted as a chair, which leads to the word "chair" in the caption). More examples are shown in Appendix D.

## 6 CONCLUSION

This paper analyzes the semantics required for the image captioning task from the perspective of human learning. We analyze the correlation between different visual tasks, then combine a variety of them and train them together, maximizing the shared semantics between all tasks. In addition, we conduct in-depth discussions on data augmentation techniques and optimizer modes to design training flow from a semantic perspective and reduce the impact of semantic errors. Experimental results show that our model is much lighter than other multi-task models and achieves great results on all tasks. Furthermore, under the multi-task joint learning architecture, by sharing semantics and learning rate, we enable image captioning tasks to achieve competitive performance without using any pre-trained models.

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

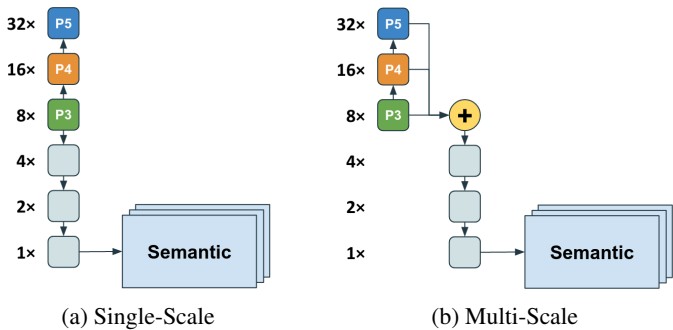

(a) Single-Scale          (b) Multi-Scale

Figure 7: **Our two types of semantic approaches.** Single-scale simply up-samples the highest resolution features, while multi-scale needs to combine all features and then up-sample.

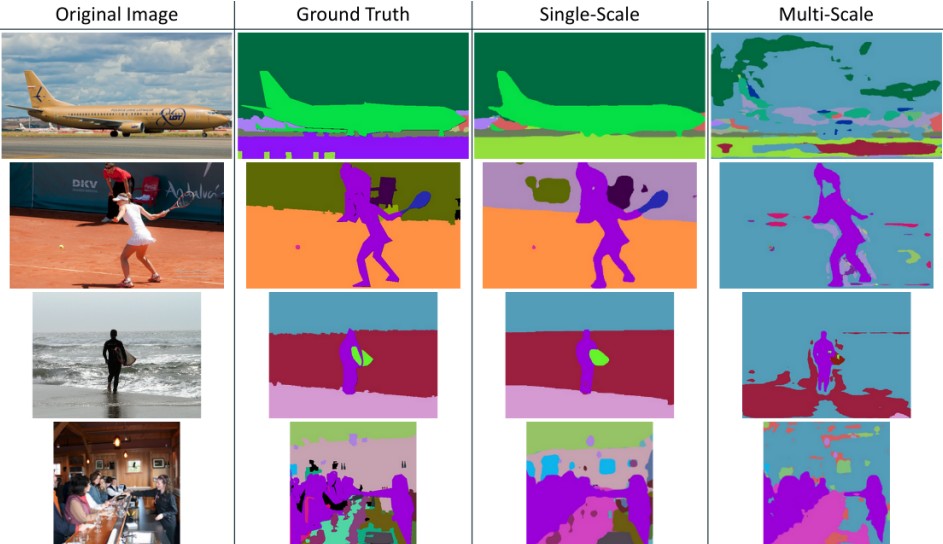

Figure 8: **Visual comparison of the results for single-scale and multi-scale semantic segmentation.** Semantic segmentation tends to be more sensitive to spatial relationships. The multi-scale approach combines low-resolution features, which results in noisy mask predictions.

## A  DESIGNED SEMANTIC APPROACHES

The semantic approaches we designed are shown in Figure 7, and the results obtained by the two approaches are shown in Figure 8. It can be seen that the prediction results of multi-scale model in the "stuff" area are noisy. We believe this is due to the semantic gap between the object detection and semantic segmentation task. Object detection attempts to classify all areas that are non-instance as background, so we can see that whatever background area is in the stuff category—sky, wall, ceiling or ground—is easily classified into the same category.

## B  ARCHITECTURE OF DATA AUGMENTATION PIPELINES

Our designed architecture of data augmentation pipelines is shown in Figure 9.

## C  VISUALIZATION OF ABLATION STUDY

From Figure 10, we found that semantic segmentation cannot distinguish different objects of the same category, and all foreground objects are treated as the same category for stuff (i.e. other).

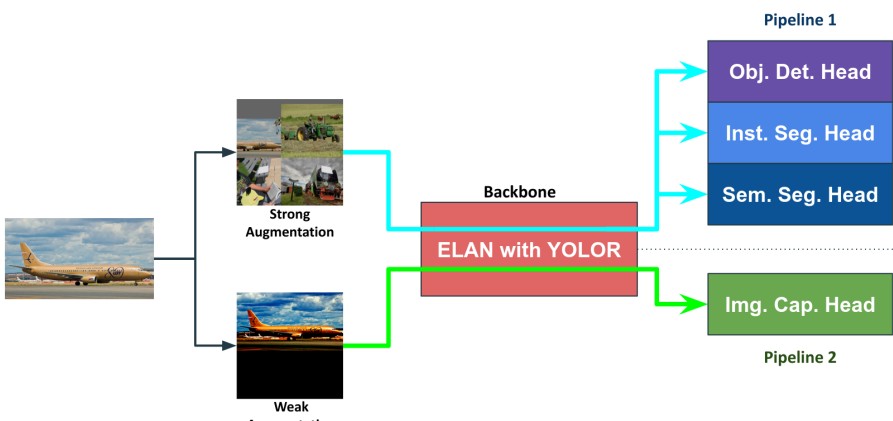

Figure 9: **The data augmentation pipelines designed for different tasks.** In order to avoid data augmentation causing the semantic change of the target task, we compose different data augmentation pipelines and use them on the corresponding target tasks respectively (e.g., strong augmentation for visual tasks, and weak augmentation for VL task).

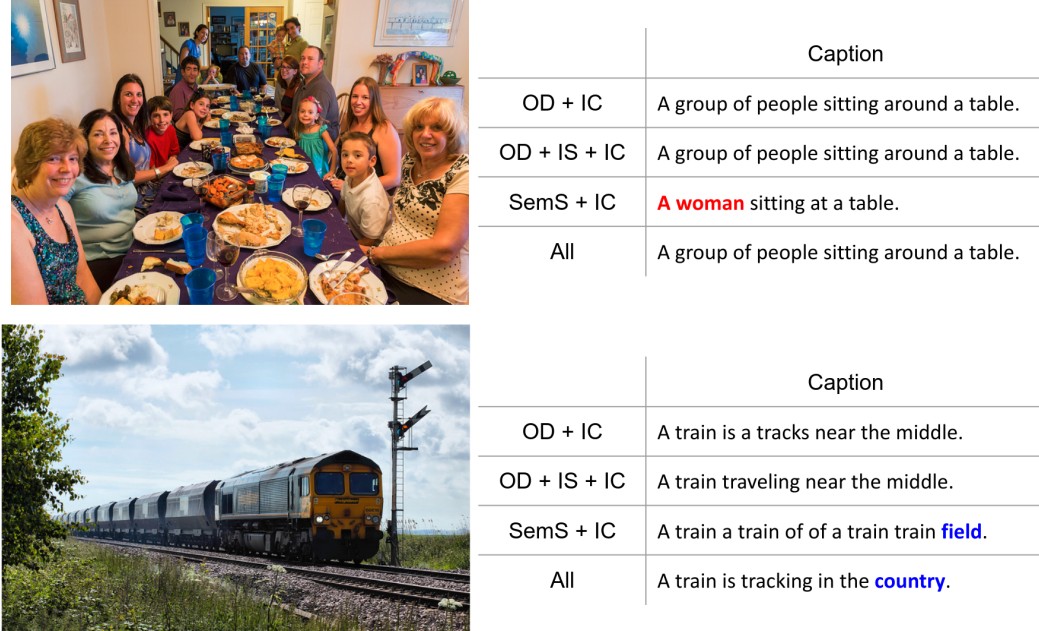

Figure 10: **Captioning results obtained by combining different visual tasks with the image captioning task.** Although semantic segmentation paired with image captioning has the worst results and cannot distinguish between different individuals of the same category at all, it can provide background semantics.

Thus, semantic segmentation (stuff segmentation) can only provide a small part of the background semantics for the image captioning task.

## D  MORE VISUALIZATION OF EXPERIMENTAL RESULTS

Figure 11 shows an example for simpler images. Our model performs well on all tasks for this batch of images. Figure 12 shows the results obtained for more complex images, and these experiments are also prone to incorrect results. For example, in the experiments of the combination of object

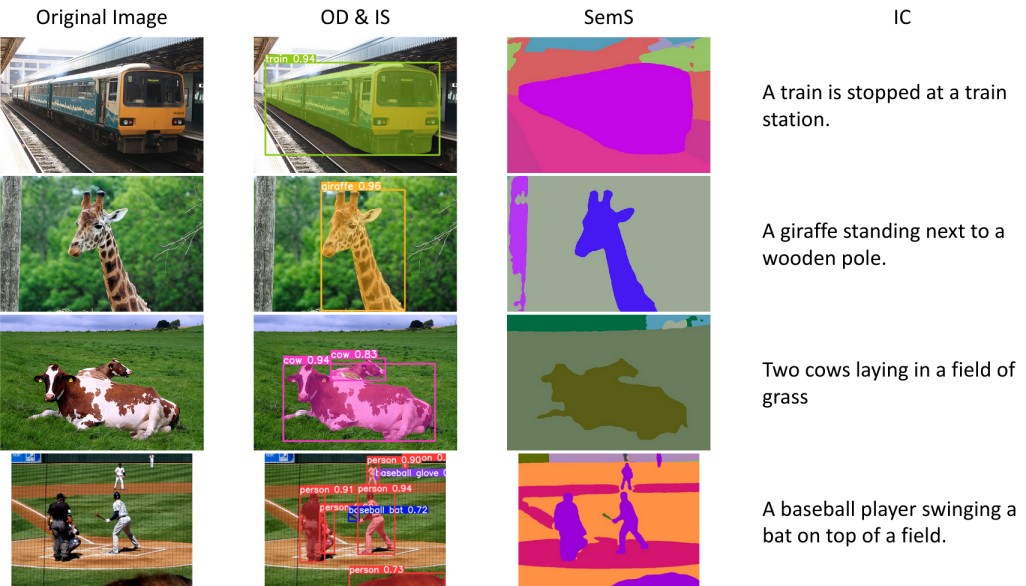

Figure 11: **Some results generated from simple images.**

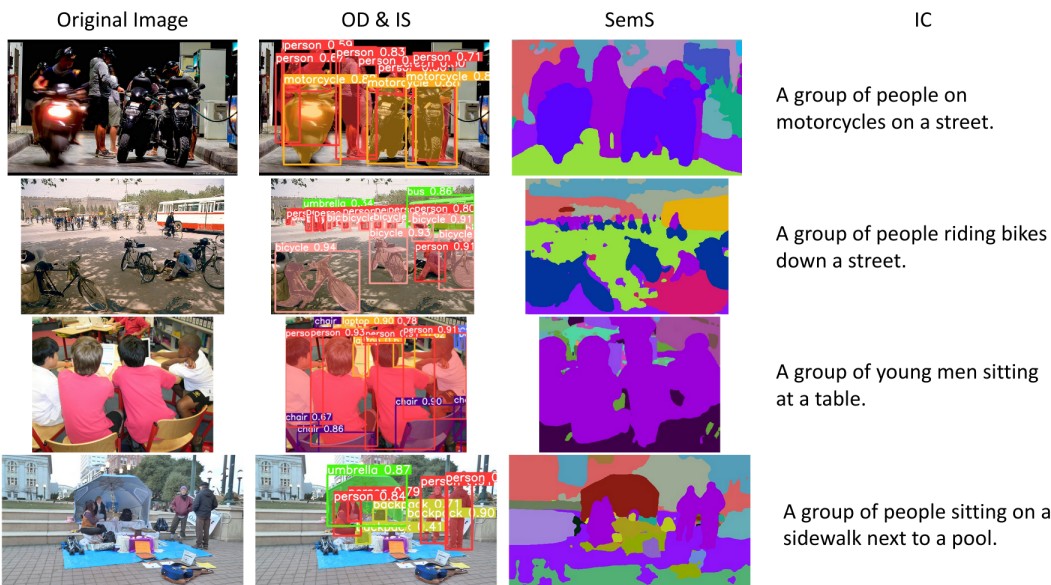

Figure 12: **Some results generated from complex images.**

detection and instance segmentation, some objects that are not easy to judge will result in inaccurate predictions, while in the experiment of semantic segmentation, messy masks will be generated due to complex spatial information, such as objects, illumination and shadow. Additionally, from the demonstrated results, most captions start with the article "a" or "an", which is a characteristic of the training data of MS COCO 2017 dataset (Lin et al., 2014).

## E   VIDEO RESULTS VISUALIZATION

We use a real video to verify our system, and the video clip used in our experiment is a ten second video taken at a road intersection (Videvo). We process a frame every second, and the captions

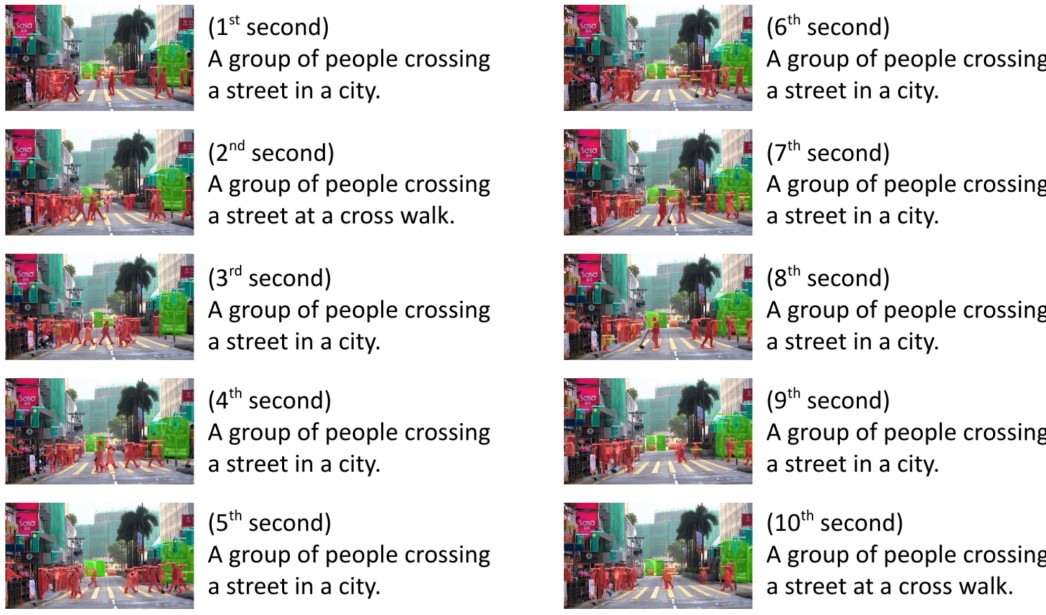

Figure 13: **Captioning results from ten second video.** One frame is captioned each second for a total of ten processed frames.

reported by the system are shown in Figure 13. It is obvious that although the content of the video changes over time, the captioning results are quite stable.

