# OpenReview forum: "YOLOR-Based Multi-Task Learning"
_ICLR.cc/2024/Conference — Submitted to ICLR 2024_

### Official Review · Reviewer_ZD4T · 2023-10-24

**Soundness:** 2 fair
**Presentation:** 3 good
**Contribution:** 1 poor
**Rating:** 3
**Confidence:** 4

**Summary:**

This paper proposes an extension of an existing model (YOLOR), training it on more tasks (including object detection, instance segmentation, semantic segmentation, and image captioning). The obtained models achieve reasonable performance compared with existing models with a compact size.

**Strengths:**

- The paper is easy to read.

- Multi-task learning for visual-textual understanding is an important and interesting topic.

**Weaknesses:**

- A significant concern raised is that the paper reads more like an engineering extension of YOLOR rather than an academic contribution with substantial technical advancements. The baseline model itself is already a multi-task learning model, and the proposal primarily focuses on designing common task heads for it.

- An essential experiment that is missing is the evaluation of the effectiveness of the multi-task models compared to their single-task counterparts. This type of comparison is widely employed in multi-task learning for computer vision [1-4]. Without conducting this experiment, it becomes challenging to accurately assess the impact and significance of multi-task learning in this context.

- YOLOR [5] should be considered the most crucial baseline and therefore should be included in the state-of-the-art (SOTA) comparison tables, such as Table 3, 4, and 5.

- The paper overlooks numerous recent works on multi-task scene understanding [1-4]. It is imperative to discuss or compare these works in the experiments to provide a comprehensive analysis of the field.

[1] MTI-Net: Multi-Scale Task Interaction Networks for Multi-Task Learning. ECCV 2020

[2] MTFormer: Multi-Task Learning via Transformer and Cross-Task Reasoning. ECCV 2022

[3] Inverted Pyramid Multi-task Transformer for Dense Scene Understanding. ECCV 2022

[4] Multi-Task Learning with Multi-query Transformer for Dense Prediction. arXiv 2022

[5] You only learn one representation: Unified network for multiple tasks. JISE 2023

**Questions:**

- Could you address my concerns above?

---

### Official Review · Reviewer_MQQ6 · 2023-10-27

**Soundness:** 3 good
**Presentation:** 3 good
**Contribution:** 2 fair
**Rating:** 5
**Confidence:** 4

**Summary:**

This paper proposes a multi-task learning framework based on YOLOR (You Only Learn One Representation). In contrast with YOLOR and YOLOv7 which only train on two vision tasks, the authors simultaneously train on four vision tasks that are chosen by human heuristics to maximize the shared semantics among tasks. To reduce the conflicts among tasks, the authors analyze data augmentation techniques and design training flow. Extensive experiments show that all tasks improve via joint training, and meanwhile maintaining a low parameter count.

**Strengths:**

1. The paper is well written and easy to follow.
2. For the first time the author successfully train on four vision tasks with one network and achieve strong performance, indicating semantically-related tasks can enhance each other.
3. The author show that not all data augmentation methods are beneficial for a specific task, and they select different augmentations for different tasks, following the principle "after changing the content of the original image, will the semantic information of corresponding task also be changed".
4. Previous work assigns the image encoder a smaller learning rate than the text encoder in multi-modal training. However, the authors find that in multi-task vision traninig, it is better to use the same learning rate for vision and text encoder.

**Weaknesses:**

1. The key contribution of the paper mainly comes from heuristics and experiment observations, there lack in-depth analysis or theoretical provement.
2. The authors tries on four semantic-related vision tasks, how the proposed method performs in a larger-scale setting, e.g. more than four tasks, is not discussed.

**Questions:**

1. It would be better to compare with panoptic segmentation methods (maybe adding some related tasks) to further show the effectiveness of the method.
2. How about the performance of the proposed method when the model has more parameters, i.e. wider or deeper.

---

### Official Review · Reviewer_Se2d · 2023-10-30

**Soundness:** 1 poor
**Presentation:** 1 poor
**Contribution:** 2 fair
**Rating:** 1
**Confidence:** 4

**Summary:**

This paper proposes a multi-task learning framework to jointly learn multiple vision: object detection, instance segmentation and semantic segmentation, and vision-language tasks: image captioning. The proposed model is built on top of YOLOR and ELAN. Though based in a multi-task learning setting, the paper assumes the prior knowledge of training tasks, and propose some additional task-specific designs: instance segmentation task relies on the prediction of object detection task; image captioning task relies on all the vision tasks.  Finally, the author provides all the engineering tricks and optimisation strategies and show the model can outperform some task-specific models in these tasks.

**Strengths:**

The proposed model is relatively lightweight, and requires no pre-training and can achieve performance competitive to some task-specific state-of-the-arts.

**Weaknesses:**

In general, I feel like the paper is quite rushed to finish. The major contribution is heavily relying on YOLOR, YOLOv7 and ELAN which were all published by Wang et al very recently in 2023, which is fine. But the paper itself does not provide all the necessary background knowledge for these papers, and therefore it’s very difficult to understand this paper itself without really understanding these other papers. Here are some other detailed comments.

1.	The paper did a fairly poor job in the literature review. The proposed method is heavily related to multi-task learning and vision-language models, but did not cover major advances in both fields and therefore missed quite a lot of important baselines. In MTL, it’s important to discuss some popular model design strategies such as in MTAN (Liu et al 2019), Cross-Stitch (Misra et al 2016), Adashare (Sun et al, 2019) for standard dense prediction tasks in computer vision, and more recently UnifiedIO, BeIT-3, Flamingo, for unifying vision and vision language tasks.

2.	The architectural design details are completely missing, and the visualisation is not clear. From Fig. 2, the image captioning task seems to be predicted from the P3 block only. But in section 3.1, the paper describes that all vision tasks are merged into image captioning block. In general, there’s no explanation of what each colour means in Fig. 2 and 3, and have absolutely no explanation of important design components like ProtoNet, P3/4/5. Coefficient, that being used in the network. Most texts in the Architecture section are simply confusing and possibly wrong, for example:
-	“Similar to the Transformer, we train the image encoder and text decoder together. ” Only recently proposed VLMs have these architecture designs but with no citations.
-	“This allows the model to be lightweight and efficient in addition to reducing training resource consumption. ” Compared to what design? Transformers are known to be not efficient since they require quadrative memory depending on the input length.
-	“Intuitively, it mimics human learning – explicit and subconscious (implicit). ” This sentence simply does not make sense for an academic paper.

3.	The experiment design does not cover important baselines, considering all evaluated tasks are well-designed and heavily evaluated within the community. In the image captioning task for example, the author only compared with an open-source model published on github, and did not compare or discuss any important VLMs published in the community. To name a few: BLIP ½, GIT, Flamingo, PaLI, SimVLM, CoCA all supposedly should have a very strong performance.

**Questions:**

See Weakness.

---

### Official Review · Reviewer_ojGP · 2023-11-02

**Soundness:** 2 fair
**Presentation:** 1 poor
**Contribution:** 1 poor
**Rating:** 3
**Confidence:** 4

**Summary:**

This paper proposes to extend the YOLOR network - previously trained for only for two tasks and extend it towards joint object detection, instance segmentation, semantic segmentation and image captioning. The authors also propose to combine their YOLOR network with ELAN, a novel method for optimizing the gradient path in a multi-task network. The authors also present a novel augmentation method to deal with a multi-task network that learns both vision tasks and NLP task (image captioning). Results are presented on MS COCO 2017

**Strengths:**

1. YOLOR is a powerful network and it is interesting to see manuscripts that attempt to build on it and improve its generalisation capabilities.

**Weaknesses:**

1. I don't understand the main novelty of the method. From my point of view, the authors propose simply to train more tasks than in the YOLOR paper and additionally use ELAN to help the learning process.

2. Results in Table 3, 4 and 5 do not seem impressive and I cannot see any significant gain achieved through the model.

3. The paper heavily uses YOLOR and ELAN yet both methods are not sufficiently presented in text. Consequently, it is very difficult to ascertain where the novelty lies in the presented work. How exactly are implicit and explicit representations learned in the current method?

4. Limited datasets - to prove ability of the model to generalise, I would expect other datasets than MS-COCO and COCO-Stuff to be presented. Datasets such as Youtube-VIS could be used as they include instance labels, bounding boxes, segmentation masks.

5. Limited baselines. Models that are based on Instance Segmentation such as QueryInst (https://arxiv.org/abs/2105.01928) also perform "multi-task learning" by jointly learning bounding boxes, instance labels and semantic masks. In addition, there are many more MTL baselines that could've been used.

6. Literature review is lacking.  The multi-task learning section is severely lacking of many novel multi-task papers. The definition of Soft-Parameter Sharing is also slightly wrong. Soft-Parameter Sharing is a class of methods where weights are shared/not-shared at all layers of the network/ This can be done in many ways. Normalizing the distance of parameters is just a way of doing it.

**Questions:**

See weaknesses

---

### Meta-Review · Area_Chair_C5fJ · 2023-12-05

**Metareview:**

All the ratings are negative, where the main concerns are: 1) technical novelty; 2) weak experimental results, e.g., limited baselines and datasets; 3) poor literature review; 4) missing technical details. Hence, the rejection rating is recommended.

**Justification For Why Not Higher Score:**

The rebuttal was not provided from the authors to address the concerns raised from the reviewers.

**Justification For Why Not Lower Score:**

N/A

---

### Decision · Program_Chairs · 2024-01-16

Reject